# Single Layer Lift-Off of CSAR62 for Dense Nanostructured Patterns

**DOI:** 10.3390/mi14040766

**Published:** 2023-03-29

**Authors:** Hanna Ohlin, Thomas Frisk, Ulrich Vogt

**Affiliations:** Department of Applied Physics, Biomedical and X-ray Physics, KTH Royal Institute of Technology, Albanova University Center, 106 91 Stockholm, Sweden

**Keywords:** lift-off, single layer, electron beam lithography, CSAR62, X-ray diffractive optics, zone plate, nanostructures, nanofabrication

## Abstract

Lift-off processing is a common method of pattern transfer for different nanofabrication applications. With the emergence of chemically amplified and semi-amplified resist systems, the possibilities for pattern definition via electron beam lithography has been widened. We report a reliable and simple lift-off process for dense nanostructured pattern in CSAR62. The pattern is defined in a single layer CSAR62 resist mask for gold nanostructures on silicon. The process offers a slimmed down pathway for pattern definition of dense nanostructures with varied feature size and an up to 10 nm thick gold layer. The resulting patterns from this process have been successfully used in metal assisted chemical etching applications.

## 1. Introduction

Lift-off methods are an important step for pattern transfer or etch mask definition in nanofabrication [1,2]. These can be described as processes where a substrate is covered with a resist layer for pattern definition, which undergoes a lithography or printing step. This leaves behind a defined structure in the resist. The resulting pattern is then covered with a thin layer that can be made out of different materials, commonly metals or oxides. The choice of material depends on the subsequent application for which the pattern is intended. It is after this that the main part of the lift-off process is conducted, which also gives raise to the name. In this step, the resist layer is removed (lifted-off), which leaves only the metal or oxide structures behind on the substrate. This makes lift-off processes a useful step in micro or nanofabrication where a high precision definition of a pattern is required. Processes like these can be of a complicated nature with many different layers to enable the desired end result [1,3]. For a simplified approach, single layer lift-off processes have been investigated in an attempt to streamline printing and pattern transfer without loss of quality or pattern integrity. In general, single-layer systems are attractive as they, when conducted successfully, make for a relatively easy process with comparatively few steps.

A common method of pattern definition is electron beam lithography (EBL), which is suitable for defining very fine structured patterns or patterns with a high demand on precision and small feature size. For EBL systems, single lift-off processes have previously been made with the resists Poly(methyl methacrylate) (PMMA) [4,5] or Hydrogen silsesquioxane (HSQ) [6]. Recent developments in resist technology has lead to chemically amplified (CAR) and chemically semi-amplified (CSAR) resists being used for EBL applications and in pattern transfer processing [7]. These resists have been suggested as good options for processes where a high dry etching resistance and sensitivity is desired in systems where high resolution is needed [8]. They offer short write times for nanostructures in both thin and thicker layers for different applications [9,10]. After pattern development, a metal layer can be deposited to the printed structures before the lift-off process itself is conducted to remove the resist layer. This leaves behind a metal pattern on the substrate which matches the exposed pattern.

Here we describe in detail a lift-off process using the chemically semi-amplified resist CSAR62 (Allresist GmbH, Germany) for the fabrication of narrow as well as thin gold zone plate nanostructures. Zone plates make for ideal test patterns, as they show a wide range of feature sizes. Both micron sized structures and sub 50 nm structures are printed and evaluated in the same experiment, and the lift-off process has to be versatile enough to facilitate processing for both. The patterns are dense in nature and have a smallest feature size of 30 nm in the outermost zone. Through the use of a single layer resist mask combined with a 50 kV EBL system, it is possible to obtain very fine structures in the desired metal on a silicon wafer [10]. A possible application of the structures is as catalyst layer for metal-assisted chemical etching (MACE).

## 2. Materials and Methods

### 2.1. Pattern Definition

For this study, Fresnel zone plates were used as an example of dense nanostructured patterns. The widths of the zones ranged from micron-sized in the centre towards 30 nm at the outermost zone. Each zone is split in sections as the final gold structures are supposed to be used as catalyst in metal assisted chemical etching [11]. Thus, the pattern consisted of densely packed segments of a wide range of sizes. An example is shown in Figure 1.

Each zone plate pattern is 150 µm in diameter and nine zone plates are printed on each chip together with supporting structures for identification and future dicing. The designed structures’ parameters were defined in MATLAB code generating the input file for the pattern generator in the electron beam lithography system.

### 2.2. Sample Preparation and E-Beam Lithography

Silicon chips were diced to 0.5–1 cm^2^ pieces before they were cleaned by sonication in isopropanol (IPA) and blown dry under N_2_ gas flow. A 5 min reactive ion-etch step (RIE, Oxford Instruments) followed in preparation for resist application. All chips were then spin coated in CSAR-6200.9 (Allresist GmbH, Strausburg, Germany), 7000 RPM for 1 min before baked on a hotplate at 150 °C for 1 min. The resulting resist layer measured approximately 90 nm in thickness.

Structures were then printed in a Voyager (Raith GmbH, Germany) 50 kV system at 120 µC/cm^2^. The printed structures were developed in amyl acetate (AR546 AllResist GmbH, Strausburg, Germany) for 1 min before the development process was stopped through a 10 s IPA soak. After a 20 s subsequent soak in penthane, the developed patterns were allowed to air dry.

The samples were coated with a 3 nm Ti adhesion layer and a 10 nm Au layer to facilitate future etching experiments. This was done through a physical vapour deposition process (PVD) utilising UHV E-gun (Thermionics, UK) deposition in-house built instrument.

### 2.3. Lift-Off

The lift-off process was adapted from a previously described method utilized for similar applications [11]. The resulting process after adaptation, from start to finish, is shown as a schematic in Figure 2.

An initial soak in AR 600-71 (Allresist GmbH, Strausburg, Germany) was conducted as soon as the Ti and Au layers had been deposited. Each sample was left in 20 mL of the AR 600-71 remover solution for 2 days at room temperature. Samples were rinsed in the soak remover solution and manually agitated vigorously to remove any larger resist residues. Samples were then transferred to individual crystallisation dishes for Cycle 1 of the lift-off process. Each step was conducted in a sonication bath at 80 kHz at a slightly elevated temperature of 30 °C. The different cycles are described in Table 1.

Samples were then removed from the DI-water and gently blown dry under nitrogen gas flow.

### 2.4. Analysis

Samples were imaged utilising scanning electron microscopy (SEM, FEI Nova 200, Altrincham, UK) with 10 kV at 5.4 mm working distance.

## 3. Results and Discussions

Successful and failed lift-off results are shown in Figure 3. In (a) and (b) structures free of resist can be seen with a good pattern integrity, very similar to the intended patterns in Figure 1. For future use of the structures in MACE, it is important that there is no resist left between the gold structures. With a failed lift-off process, the pattern will have residual resist, as shown in (c) and (d) of the same Figure 3.

The process itself has been adapted from previously used methods [11]. With this as a starting point, care was taken to improve the process in such a fashion that the yield of usable structures post lift-off would be as high as possible. After an iterative process of changes, this resulted in the implementation of a few major process step redesigns.

First and foremost, the remover solution used was changed from dimethylsuccinate to remover AR600-71. The main component of this remover solution is dioxolane, which offers a higher solubility for the CSAR62 resist, even when the film has been baked at relatively high temperatures. To further facilitate the solubility of the resist, the baking temperature before exposure was lowered from 180 °C to 150 °C. The over-night soak was extended from one to two days, as this showed significant improvement in loosening the resist from the silicon. Also, the suggested lift-off process now consists of several cycles to further increase the amount of exposure the resist receives from the remover solution.

Two different sonication frequencies were tested. The starting point from Akan et al. [11] suggested 40 MHz, which offered partial success in removing the resist in a satisfactory fashion. In an attempt to increase the yield, a higher frequency of 80 MHz was tested and found to give better results. This could be a first indication that there is a frequency dependency to the lift-off process, but future studies into this matter are needed for a better insight on how to more readily adapt single layer processes for similar structures.

Comparing the process described by Akan el al. [11] to the cyclic process used in this study, there is also a difference in processing temperature. This is another side-effect caused by the change of remover solution, as the boiling point of dioxolane (74 °C) is significantly lower than that of dimethylsuccinate (196 °C). The suggested processing temperature for AR600-71 is no higher than room temperature. Whilst an elevated temperature might improve the mobility and solubility alike, caution was preferred in this case. Thus, only a slight elevation to 30 °C was introduced.

The biggest challenge is to complete the lift-off process for the smallest structures, as these seem to be more prone to failure than the larger features towards the middle. After two nights of soaking, cycle 1 and 2 will be efficient in removing the bulk of the resist, but not all. Thus, cycle 3 and 4 were added so to ensure any remaining flakes and areas in the more densely structured parts of the pattern were cleared. Cycle 5 and 6 are the only steps that remains relatively unchanged in comparison to the previously used method, since they do not add to any efficiency of the lift-off process itself, only acting as a cleaning step for following processing.

In Figure 4, possible results of the lift-off process for the smallest structures are shown in detail. Judging by the images in (a) and (b), it looks like dissolving the resist is harder than anticipated. Residual resist seems to be loosening in long ribbons rather than tiny flakes, suggesting that CSAR62 in this state is rather stable. That a clean and satisfactory lift-off process can be achieved is shown in (c) and (d), where no residual resist ribbons are present between the structures, even at their most narrow and dense point. This adds merit to the suggested cyclic process, as it seems to be important for a complete lift-off. When the first round of resist is stripped in cycle 1 and 2, there will be new sites where the remover can interact with the remaining flakes of the resist, which makes cycle 3 and 4 critical for success. It can be debated whether further improvements to the yield could be obtained from adding another soak between cycle 2 and 3. This was however deemed as too time consuming, but could certainly be interesting for future investigations.

The resist applied measured approximately 90 nm in thickness. With 10 nm gold deposited at both the bottom of the narrow trenches and the trench wall tops, a narrow window of 80 nm is all that is left exposed to the solvents on the sidewalls of the trenches. A lift-off process with a 15 nm gold layer was attempted, but failed. This could be due to the metal-free resist openings being too small, or due to the (thin) metal layer deposited on resist sidewalls. Maintaining an acceptable ratio of exposed resist seems critical for the lift-off process to work, as no residual resist can be tolerated for the subsequent processing steps. A thicker layer of resist would be ideal, as this would enable a thicker layer of gold to be deposited. Such layers are expected to further facilitate the MACE process [12]. This causes a conflict between different optimisation steps outside the actual lift-off process. The thicker the gold pattern, the better it will serve the following MACE-processing. However, a thick gold layer will also cover the resist to a larger extent, limiting access to sites where the remover solution can interact chmically with the resist.

The main advantage of CSAR62 is it’s high sensitivity compared to resists such as ZEP or PMMA [7,9], offering faster write times. There have been comparatively few studies investigating which resist type would be optimal for a lift-off process, and especially what constitutes the optimal resist-to-metal aspect ratio for a successful lift-off. Thicker resist layers seem to offer a possibility for thicker metal layers [5]. For thinner resist layers options such as nanoparticle lift-off methods have been suggested for the fabrication of gold nanostructures [4]. There seems to be a limit to the smallest possible metal-resist aspect ratio, with undercut processing and bilayer approaches as possible ways to maximise this ratio [13].

It can be discussed whether adding a lift-off resist (LOR) layer could facilitate the process. This could potentially increase the yield of samples with a successful pattern transfer. Presently, no LOR-layer matrices for CSAR62 are offered by the supplier. Further studies on both single layer and bi-layer lift-off processes with CSAR62 could offer insight on how to make an even more reliable method.

## 4. Conclusions

We report a successful single layer lift-off process for dense nanostructured patterns in CSAR-62. The process has been used numerous times in preparation of samples with gold nanostructures for metal assisted chemical etching demanding clean patterns with good pattern definition. The resist layer is stripped by a cyclic solvent exchange procedure that combines repeated dissolution of the resist helped by sonication at high frequencies. The deposited gold is used to define the pattern on the silicon in a satisfying, repeatable and reliable fashion. The process is well suited for patterns and processes demanding small feature size structures, but can also be used for larger feature sizes in the micrometer regime.

## Figures and Tables

**Figure 1 micromachines-14-00766-f001:**
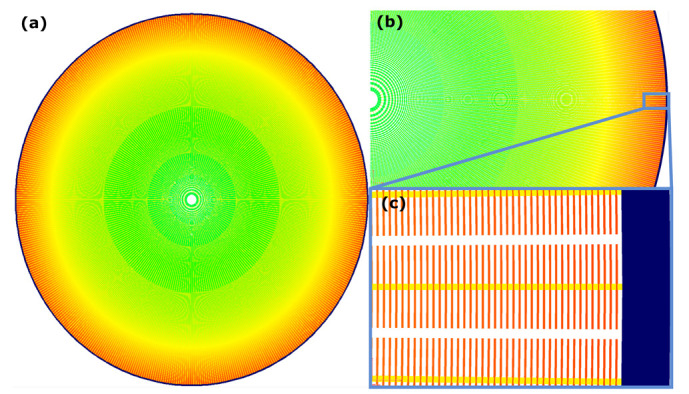
A rendered figure of the pattern intended for printing. The different colours indicate the different doses of exposure utilised during printing, where blue-green colours mark low doses and orange-red higher doses. In (**a**) a full zone plate pattern render is depicted. Feature sizes shrink radially whilst dose is increasing radially. In (**b**) a close-up of the zone plate is shown, whilst (**c**) shows the very outermost zones to be printed, where the feature size reaches its smallest value at 30 nm. These are shown in red, as the dose is comparatively high in these areas. The area enlarged between figure (**b**,**c**) is framed in light blue. The very outermost area in dark blue shows a supporting structure, a proximity ring printed with low dose, and with a width of 500 nm.

**Figure 2 micromachines-14-00766-f002:**
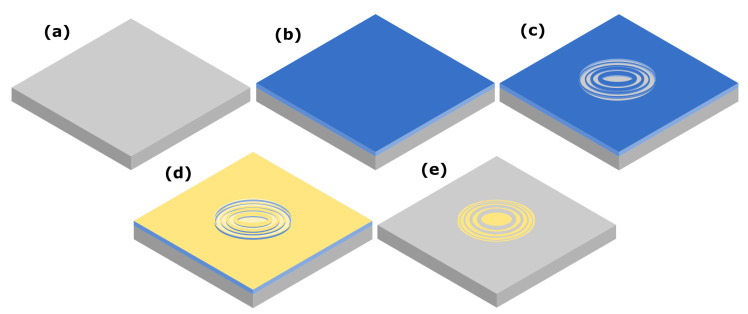
A schematic of the patterning and lift-off process, resulting in a transferred gold pattern to the raw silicon beneath. In (**a**), the starting material is depicted, showing the raw Si substrate. Progressing to (**b**) shows the sample covered with a thin layer of resist by spin coating. In (**c**) the result of the E-beam lithography pattern definition is shown, after development. A metalisation step follows (**d**) where the developed pattern is coated with a thin layer of gold. Lastly, the result of the lift-off procedure itself is shown in (**e**), which which leaves a finished gold zone-plate pattern on the silicon substrate.

**Figure 3 micromachines-14-00766-f003:**
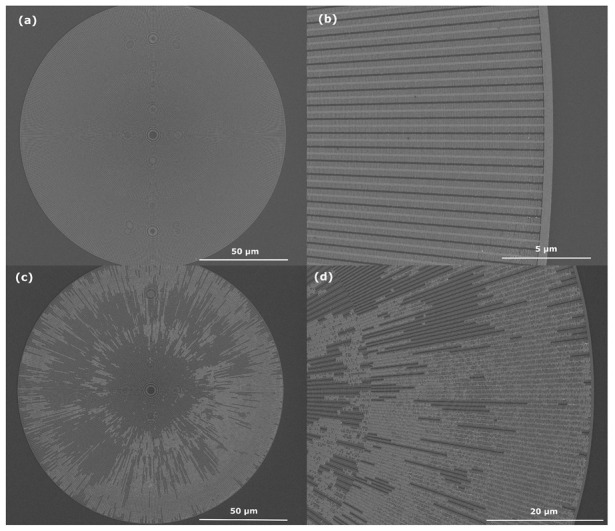
(**a**,**b**) Zone plate gold pattern on silicon after a clean lift-off with no visible resist residuals. The lighter areas are gold deposited on the silicon, whilst the darker areas are silicon with no gold deposited. The pattern is of a “fishbone” design with outermost features of 30 nm width. As shown, no resist residuals have been left behind in the pattern. (**c**,**d**) shows a zone plate pattern after a failed lift-off process. Large parts of the pattern is still covered in gold-coated resist, which makes further use of the pattern impossible.

**Figure 4 micromachines-14-00766-f004:**
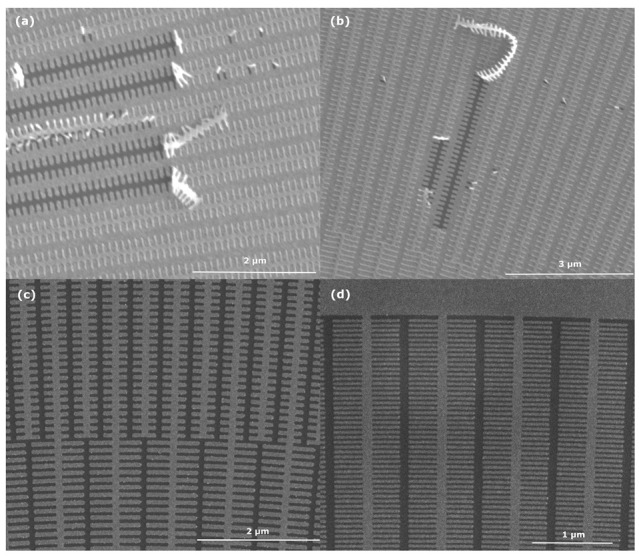
A comparison between failed and successful lift-off processes with dense, small structures. (**a**,**b**) that were processed without the resulting cyclic lift-off process shows sporadic lift-off where only moderate amount of resist has departed from the silicon chip, behaving in a zipper-like fashion. In (**c**,**d**) which have been processed with the revised process the completely cleared pattern is shown for dense 1:1 structures as small as 30 nm in width.

**Table 1 micromachines-14-00766-t001:** Description of the cycled steps of the lift-off process.

Cycle	Chemical	Time (min)
Cycle 1	Acetone	10 min
Cycle 2	AR600-71	10 min
Cycle 3	Acetone	5 min
Cycle 4	AR600-71	5 min
Cycle 5	IPA	10 min
Cycle 6	DI	10 min

## Data Availability

Data available on request.

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
