# Peer review of "Single Layer Lift-Off of CSAR62 for Dense Nanostructured Patterns"

_micromachines, 2023, doi:10.3390/mi14040766_

Round 1

Reviewer 1 Report

The manuscript “Single Layer Lift-Off of CSAR62 for Dense Nanostructured Patterns” proposed a reliable and simple lift-off process for dense nanostructured pattern in CSAR62. Related experiments and results are shown and illustrated in detail. However, there are some problems including content and form of the paper. Hereafter lists a few questions about the experiment.

1. The content of the article is relatively complete, but there are some questions that are not very clear. For example, one of the key words "Lift-off". Although you mentioned it in an earlier paper, this key word plays a key role in the article and you should briefly define it again to let the reader know what the word means in this paper.

2. The explanation should be more substantive, and the sentence should be more logical. For example, in the results and discussions, “the remover solution used was changed from dimethylsuccinate, as the efficiency of AR600-91 was suggested to be the better choice by the manufacturer”. Why is it better to use AR600-91, just by manufacturer's suggestion? Specific mechanism analysis should be given.

3. For the sonication frequency, only two frequencies of 40 MHz and 80MHz are not enough to show that the higher the frequency, the better the result. Please supply more experiments with several frequencies for analysis and as a proof.

4. In Fig. 1, The description is a little fuzzy, it is suggested to enlarge Fig. 1 (b) using the selected area in the red box to get Fig. 1 (c).

5. The normative problem of the text description of figures. In Fig. 2, you put parentheses at the end of each sentence which is different from the other figures. Please modify it.

6. Please pay attention to the rigor of figures. The white lines of the scale on the four pictures in Fig. 3 and Fig. 4 should be the same length, not too long in this image and too shorter in the next.

7. Your article is related to the following work. It is suggested to cite the following papers in the references:Superhydrophobic surfaces based on ZnO-constructed hierarchical architectures [J]. Microelectronic Engineering. 2015, 141: 44-50 

Author Response

Comment 1: "The content of the article is relatively complete, but there are some questions that are not very clear. For example, one of the key words ”Lift-off”. Although you mentioned it in an earlier paper, this key word plays a key role in the article and you should briefly define it again to let the reader know what the word means in this paper."

We have expanded our explanation on the terminology of lift-off procession and elaborated in the introduction to make this clearer.

Comment 2: "The explanation should be more substantive, and the sentence should be more logical. For example, in the results and discussions, “the remover solution used was changed from dimethylsuccinate, as the efficiency of AR600-91 was suggested to be the better choice by the manufacturer”. Why is it better to use AR600-91, just by manufacturer’s suggestion? Specific mechanism analysis should be given."

We have amended the method section with a better description of the remover, as well as amended the discussion accordingly to provide a more in-going elaboration.

Comment 3: "For the sonication frequency, only two frequencies of 40 MHz and 80MHz are not enough to show that the higher the frequency, the better the result. Please supply more experiments with several frequencies for analysis and as a proof."

The purpose of the paper is to offer a reliable method to perform a single-layer lift-off of CSAR62. Whilst we certainly understand the value of an extended study on frequencies in the sonication process, we believe that it would be outside the scope of this paper, and better fit for a separate study. Thus, we have reworded the section on sonication so to clarify the purpose.

Comment 4: "In Fig. 1, The description is a little fuzzy, it is suggested to enlarge Fig. 1 (b) using the selected area in the red box to get Fig. 1 (c)."

The description has been updated to better explain what is depicted. The figure itself has also been updated with coloured frames to clarify what areas are depicted.

Comment 5: "The normative problem of the text description of figures. In Fig. 2, you put parentheses at the end of each sentence which is different from the other figures. Please modify it."

We have rewritten the figure caption so it is more consistent with those of the other figures in the paper.

Comment 6: "Please pay attention to the rigor of figures. The white lines of the scale on the four pictures in Fig. 3 and Fig. 4 should be the same length, not too long in this image and too shorter in the next."

The scale bars are not congruent with one another due to the choice of pictures. Attention has been paid so to choose images in such a way that each series of images will highlight smaller and smaller features. Thus, we have chosen to display the scale bars as made by the instrument so to assure a correct measure in the pictures over making sure they are of the same length.

Comment 7: "Your article is related to the following work. It is suggested to cite the following papers in the references: Superhydrophobic surfaces based on ZnO-constructed hierarchical architectures J. Microelectronic Engineering. 2015, 141: 44-50."

We have reviewed the suggested research offered by the reviewer and believe that it might be a bit too far away from the topic in our manuscript. Thus, whilst the suggestion on an expanded list of references is appreciated, we have chosen to exclude this as it does not connect to the main theme of our paper, lift-off processes.

For full response to reviewers and editor, please see the attached file.

Reviewer 2 Report

The article describes a successful single layer lift-off process for fabricating dense nanostructured patterns using e-beam lithography and CSAR-62 resist. The process offers a streamlined and efficient approach to pattern transfer with high resolution and fidelity, and is well-suited for structures demanding small feature sizes and high dry etching resistance. The authors report successful fabrication of 10 nm gold nanostructures for metal-assisted chemical etching, with the process being repeatable, reliable, and providing clean pattern definition. Here are my questions regarding this manuscript.

1.     Could the authors consider using a LOR layer beneath the CSAR-62 resist to facilitate the lift-off process, given that single layer resists are known to be more challenging in lift-off processes compared to double layer lift-off processes? Incorporating a LOR layer can create a sacrificial layer that can be easily removed during the lift-off process, potentially reducing the risk of damage to the underlying patterned layer and improving the efficiency of the process.

2.     Would it be possible for the authors to provide a cross-sectional SEM image of the resist pattern after development? This image would be useful in characterizing the sidewall profile and potentially help to determine why the 15 nm gold film failed in the lift-off process. It would also provide valuable information regarding the quality of the pattern and the overall success of the lift-off process.

3.     Have you explored the possibility of raising the temperature of the remover used in the lift-off process to assist with the removal of the resist layer?

Author Response

Comment 1: "Could the authors consider using a LOR layer beneath the CSAR-62 resist to facilitate the lift-off process, given that single layer resists are known to be more challenging in lift-off processes compared to double layer lift-off processes? Incorporating a LOR layer can create a sacrificial layer that can be easily removed during the lift-off process, potentially reducing the risk of damage to the underlying patterned layer and improving the efficiency of the process."

To the extent of our knowledge, there are no formulas fit for LOR in the case of CSAR62. We do however agree with the reviewer, it would likely be a helpful step in process, especially given the fact that we presently work with very thin layers of resist so to facilitate the lithography. We have added a comment on this in the discussion to clarify the choice of processing path.

Comment 2: "Would it be possible for the authors to provide a cross-sectional SEM image of the resist pattern after development? This image would be useful in characterizing the sidewall profile and potentially help to determine why the 15 nm gold film failed in the lift-off process. It would also provide valuable information regarding the quality of the pattern and the overall success of the lift-off process."

Whilst the suggestion of the reviewer is certainly agreeable, our experience is that it is difficult to image our resist structures with SEM. This is due to the fact that the resist cannot withstand the electron beam for any prolonged times. Had the structures been somewhat larger, this might have been easier, but with dense and very fine nanostructures like these, cross section images are hard to obtain. Thus we chose to omit that from our manuscript, as the intent was more to present a successful process that could potentially be of use for others seeking to create similar structures.

Comment 3: "Have you explored the possibility of raising the temperature of the remover used in the lift-off process to assist with the removal of the resist layer?"

As the described process was adapted from a previously utilized method (Akan et. al 2020), this was absolutely considered. Initially, we raised the temperature of the primary remover solution, dimethylsuccinate, but found that this still did not yield satisfactory results. Thus, the remover solution was replaced by AR600-71 from the same manufacturer. Whilst this one is deemed to be more efficient even at room temperature, it was lightly heated to further increase the solution mobility. AR600-71 is however more temperature sensitive, which does not allow us to heat it past 30-40 degrees. We have amended the method section to clarify this point.

For the full resubmission letter to reviewers and editor, please see the attached file.

Round 2

Reviewer 1 Report

I have no further comments to the paper. I suggest the paper can be accepted.

Thanks,